# Unifying machine learning and quantum chemistry with a deep neural network for molecular wavefunctions

K.T. Schütt [1], M. Gastegger[1], A. Tkatchenko[2]*, K.-R. Müller[1,3,4]* & R.J. Maurer[5]*

Machine learning advances chemistry and materials science by enabling large-scale exploration of chemical space based on quantum chemical calculations. While these models supply fast and accurate predictions of atomistic chemical properties, they do not explicitly capture the electronic degrees of freedom of a molecule, which limits their applicability for reactive chemistry and chemical analysis. Here we present a deep learning framework for the prediction of the quantum mechanical wavefunction in a local basis of atomic orbitals from which all other ground-state properties can be derived. This approach retains full access to the electronic structure via the wavefunction at force-field-like efficiency and captures quantum mechanics in an analytically differentiable representation. On several examples, we demonstrate that this opens promising avenues to perform inverse design of molecular structures for targeting electronic property optimisation and a clear path towards increased synergy of machine learning and quantum chemistry.

---

[1] Machine Learning Group, Technische Universität Berlin, 10587 Berlin, Germany. [2] Physics and Materials Science Research Unit, University of Luxembourg, L-1511 Luxembourg, Luxembourg. [3] Department of Brain and Cognitive Engineering, Korea University, Anam-dong, Seongbuk-gu, Seoul 02841, Korea. [4] Max-Planck-Institut für Informatik, Saarbrücken, Germany. [5] Department of Chemistry, University of Warwick, Gibbet Hill Road, CV4 7AL Coventry, UK. *email: alexandre.tkatchenko@uni.lu; klaus-robert.mueller@tu-berlin.de; r.maurer@warwick.ac.uk

Machine learning (ML) methods reach ever deeper into quantum chemistry and materials simulation, delivering predictive models of interatomic potential energy surfaces[1-6], molecular forces[7,8], electron densities[9], density functionals[10], and molecular response properties such as polarisabilities[11], and infrared spectra[12]. Large data sets of molecular properties calculated from quantum chemistry or measured from experiment are equally being used to construct predictive models to explore the vast chemical compound space[13-17] to find new sustainable catalyst materials[18], and to design new synthetic pathways[19]. Recent research has explored the potential role of machine learning in constructing approximate quantum chemical methods[20], as well as predicting MP2 and coupled cluster energies from Hartree–Fock orbitals[21,22]. There have also been approaches that use neural networks as a basis representation of the wavefunction[23-25].

Most existing ML models have in common that they learn from quantum chemistry to describe molecular properties as scalar, vector, or tensor fields[26,27]. Figure 1a shows schematically how quantum chemistry data of different electronic properties, such as energies or dipole moments, is used to construct individual ML models for the respective properties. This allows for the efficient exploration of chemical space with respect to these properties. Yet, these ML models do not explicitly capture the electronic degrees of freedom in molecules that lie at the heart of quantum chemistry. All chemical concepts and physical molecular properties are determined by the electronic Schrödinger equation and derive from the ground-state wavefunction. Thus, an electronic structure ML model that directly predicts the ground-state wavefunction (see Fig. 1b) would not only allow to obtain all ground-state properties, but could open avenues towards new approximate quantum chemistry methods based on an interface between ML and quantum chemistry. Hegde and Bowen[28] have explored this idea using kernel ridge regression to predict the band structure and ballistic transmission in a limited

study on straining single-species bulk systems with up to μfour atomic orbitals. Another recent example of this scheme is the prediction of coupled-cluster singles and doubles amplitudes from MP2-derived properties by Townsend and Vogiatzis[29].

In this work, we develop a deep learning framework that provides an accurate ML model of molecular electronic structure via a direct representation of the electronic Hamiltonian in a local basis representation. The model provides a seamless interface between quantum mechanics and ML by predicting the eigenvalue spectrum and molecular orbitals (MOs) of the Hamiltonian for organic molecules close to 'chemical accuracy' (~0.04 eV). This is achieved by training a flexible ML model to capture the chemical environment of atoms in molecules and of pairs of atoms. Thereby, it provides access to electronic properties that are important for chemical interpretation of reactions such as charge populations, bond orders, as well as dipole and quadrupole moments without the need of specialised ML models for each property. We demonstrate how our model retains the conceptual strength of quantum chemistry by performing an ML-driven molecular dynamics simulation of malondialdehyde showing the evolution of the electronic structure during a proton transfer while reducing the computational cost by 2–3 orders of magnitude. As we obtain a symmetry-adapted and analytically differentiable representation of the electronic structure, we are able to optimise electronic properties, such as the HOMO-LUMO gap, in a step towards inverse design of molecular structures. Beyond that, we show that the electronic structure predicted by our approach may serve as input to further quantum chemical calculations. For example, wavefunction restarts based on this ML model provide a significant speed-up of the self-consistent field procedure (SCF) due to a reduced number of iterations, without loss of accuracy. The latter showcases that quantum chemistry and machine learning can be used in tandem for future electronic structure methods.

## Results

**Atomic representation of molecular electronic structure.** In quantum chemistry, the wavefunction associated with the electronic Hamiltonian $\hat{H}$ is typically expressed by anti-symmetrised products of single-electron functions or molecular orbitals. These are represented in a local atomic orbital basis of spherical atomic functions $\left|\psi_m\right\rangle = \sum_i c_m^i \left|\phi_i\right\rangle$ with varying angular momentum. As a consequence, one can write the electronic Schrödinger equation in matrix form

$$\mathbf{H}\mathbf{c}_m = \epsilon_m \mathbf{S}\mathbf{c}_m, \tag{1}$$

where the Hamiltonian matrix $\mathbf{H}$ may correspond to the Fock or Kohn–Sham matrix, depending on the chosen level of theory[30]. In both cases, the Hamiltonian and overlap matrices are defined as:

$$H_{ij} = \left\langle \phi_i \middle| \hat{H} \middle| \phi_j \right\rangle \tag{2}$$

and

$$S_{ij} = \left\langle \phi_i \middle| \phi_j \right\rangle. \tag{3}$$

The eigenvalues $\epsilon_m$ and electronic wavefunction coefficients $c_m^i$ contain the same information as $\mathbf{H}$ and $\mathbf{S}$ where the electronic eigenvalues are naturally invariant to rigid molecular rotations, translations or permutation of equivalent atoms. Unfortunately, as a function of atomic coordinates and changing molecular configurations, eigenvalues and wavefunction coefficients are not well-behaved or smooth. State degeneracies and electronic level crossings provide a challenge to the direct prediction of eigenvalues and wavefunctions with ML techniques. We address this

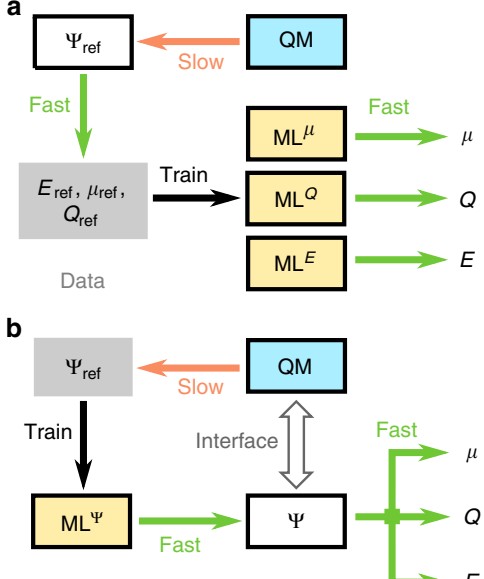

**Fig. 1** Synergy of quantum chemistry and machine learning. **a** Forward model: ML predicts chemical properties based on reference calculations. If another property is required, an additional ML model has to be trained. **b** Hybrid model: ML predicts the wavefunction. All ground state properties can be calculated and no additional ML is required. The wavefunctions can act as an interface between ML and QM

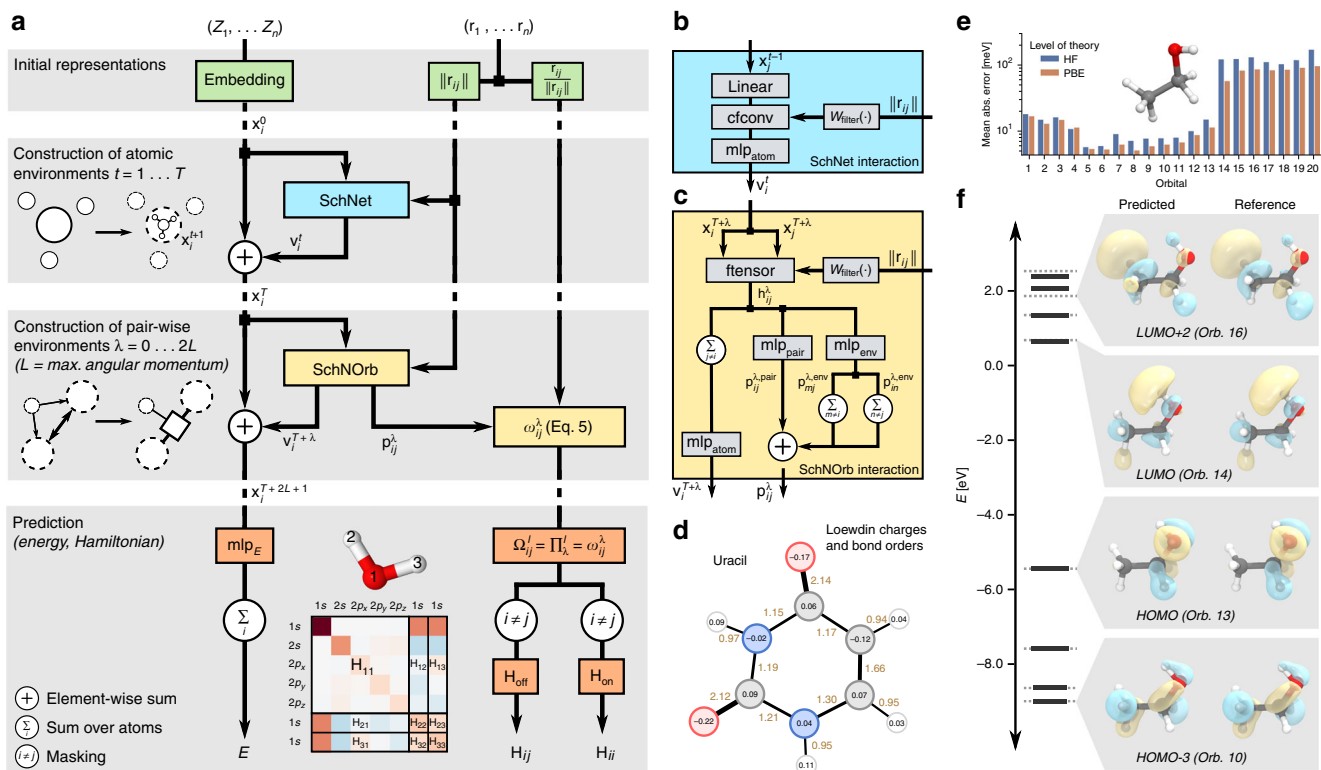

**Fig. 2** Prediction of electronic properties with SchNOrb. **a** Illustration of the network architecture. The neural network architecture consists of three steps (grey boxes) starting from initial representations of atom types and positions (top), continuing with the construction of representations of chemical environments of atoms and atom pairs (middle) before using these to predict energy and Hamiltonian matrix respectively (bottom). The left path through the network to the energy prediction $E$ is rotationally invariant by design, while the right pass to the Hamiltonian matrix **H** allows for a maximum angular momentum $L$ of predicted orbitals by employing a multiplicative construction of the basis $\boldsymbol{\omega}_{ij}$ using sequential interaction passes $l = 0...2L$. The onsite and offsite blocks of the Hamiltonian matrix are treated separately. The prediction of overlap matrix **S** is performed analogously. **b** Illustration of the SchNet interaction block[32]. **c** Illustration of SchNorb interaction block. The pairwise representation $\mathbf{h}_{ij}^l$ of atoms $i, j$ is constructed by a factorised tensor layer $f_{\text{tensor}}$ from atomic representations as well as the interatomic distance. Using this, rotationally invariant interaction refinements $\mathbf{v}_i^m$ and basis coefficients $\mathbf{p}_{ij}^l$ are computed. **d** Loewdin population analysis for uracil based on the density matrix calculated from the predicted Hamiltonian and overlap matrices. **e** Mean abs. errors of lowest 20 orbitals (13 occupied + 7 virtual) of ethanol for Hartree–Fock and DFT@PBE. **f** The predicted (solid black) and reference (dashed grey) orbital energies of an ethanol molecule for DFT. Shown are the last four occupied and first four unoccupied orbitals, including HOMO and LUMO. The associated predicted and reference molecular orbitals are compared for four selected energy levels

problem with a deep learning architecture that directly describes the Hamiltonian matrix in local atomic orbital representation.

**SchNOrb deep learning framework**. SchNOrb (SchNet for Orbitals) presents a framework that captures the electronic structure in a local representation of atomic orbitals that is common in quantum chemistry. Figure 2a gives an overview of the proposed architecture. SchNOrb extends the deep tensor neural network SchNet[31] to represent electronic wavefunctions. The core idea is to construct symmetry-adapted pairwise features $\boldsymbol{\Omega}_{ij}^l$ to represent the block of the Hamiltonian matrix corresponding to atoms $i, j$. They are written as a product of rotationally invariant ($\lambda = 0$) and covariant ($\lambda > 0$) components $\boldsymbol{\omega}_{ij}^\lambda$ which ensures that – given a sufficiently large feature space – all rotational symmetries up to angular momentum $l$ can be represented:

$$\boldsymbol{\Omega}_{ij}^l = \prod_{\lambda=0}^l \boldsymbol{\omega}_{ij}^\lambda \quad \text{with } 0 \le l \le 2L \qquad (4)$$

$$\boldsymbol{\omega}_{ij}^\lambda = \begin{cases} \mathbf{p}_{ij}^\lambda \otimes \mathbb{1}_D & \text{for } \lambda = 0 \\ \left[\mathbf{p}_{ij}^\lambda \otimes \dfrac{\mathbf{r}_{ij}}{\|\mathbf{r}_{ij}\|}\right] \mathbf{W}^\lambda & \text{for } \lambda > 0 \end{cases}, \qquad (5)$$

here, $\mathbf{r}_{ij}$ is the vector pointing from atom $i$ to atom $j$, $\mathbf{p}_{ij}^\lambda \in \mathbb{R}^B$ are rotationally invariant coefficients and $\mathbf{W}^\lambda \in \mathbb{R}^{3 \times D}$ are learnable parameters projecting the features along $D$ randomly chosen directions. This allows to rotate the different factors of $\boldsymbol{\Omega}_{ij}^l \in \mathbb{R}^{B \cdot D}$ relative to each other and further increases the flexibility of the model for $D > 3$. In case of $\lambda = 0$, the coefficients are independent of the directions due to rotational invariance.

We obtain the coefficients $\mathbf{p}_{ij}^\lambda$ from an atomistic neural network, as shown in Fig. 2a. Starting from atom type embeddings $\mathbf{x}_i^0$, rotationally invariant representations of atomistic environments $\mathbf{x}_i^T$ are computed by applying $T$ consecutive interaction refinements. These are by construction invariant with respect to rotation, translation and permutations of atoms. This part of the architecture is equivalent to the SchNet model for atomistic predictions (see refs. [32,33]). In addition, we construct representations of atom pairs $i,j$ that will enable the prediction of the

coefficients $\mathbf{p}_{ij}^{\lambda}$. This is achieved by $2L+1$ SchNOrb interaction blocks, which compute the coefficients $\mathbf{p}_{ij}^{\lambda}$ with a given angular momentum $\lambda$ with respect to the atomic environment of the respective atom pair $ij$. This corresponds to adapting the atomic orbital interaction based on the presence and position of atomic orbitals in the vicinity of the atom pair. As shown in Fig. 2b, the coefficient matrix depends on pair interactions $\text{mlp}_{\text{pair}}$ of atoms $i, j$ as well as environment interactions $\text{mlp}_{\text{env}}$ of atom pairs $(i, m)$ and $(n, j)$ for neighbouring atoms $m, n$. These are crucial to enable the model to capture the orientation of the atom pair within the molecule for which pair-wise interactions of atomic environments are not sufficient.

The Hamiltonian matrix is obtained by treating on-site and off-site blocks separately. Given a basis of atomic orbitals up to angular momentum $L$, we require pair-wise environments with angular momenta up to $2L$ to describe all Hamiltonian blocks

$$\tilde{\mathbf{H}}_{ij} = \begin{cases} \mathbf{H}_{\text{off}}\left(\left[\mathbf{\Omega}_{ij}^l\right]_{0 \leq l \leq 2L+1}\right) & \text{for } i \neq j \\ \mathbf{H}_{\text{on}}\left(\left[\mathbf{\Omega}_{im}^l\right]_{\substack{m \neq i \\ 0 \leq l \leq 2L+1}}\right) & \text{for } i = j \end{cases}, \quad (6)$$

The predicted Hamiltonian is obtained through symmetrisation $\mathbf{H} = \frac{1}{2}(\tilde{\mathbf{H}} + \tilde{\mathbf{H}}^{\mathsf{T}})$. $\mathbf{H}_{\text{off}}$ and $\mathbf{H}_{\text{on}}$ are modelled by neural networks that are described in detail in the methods section. The overlap matrix $\mathbf{S}$ can be obtained in the same manner. Based on this, the orbital energies and coefficients can be calculated according to Eq. (1). The computational cost of the diagonalisation is negligible for the molecules and basis sets we study here (<1 ms). For large basis sets and molecules, when the diagonalisation starts to dominate the computational cost, our method requires only a single diagonalization instead of one per SCF step. In addition to the Hamiltonian and overlap matrices, we predict the total energy separately as a sum over atom-wise energy contributions, in analogy with the conventional SchNet treatment[32] to drive the molecular dynamics simulations.

**Learning electronic structure and derived properties**. The proposed SchNOrb architecture allows us to perform predictions of total energies, Hamiltonian and overlap matrices in end-to-end fashion using a combined regression loss. We train separate neural networks for several data sets of water as well as ethanol, malondialdehyde, and uracil from the MD17 dataset[7]. The reference calculations were performed with Hartree-Fock (HF) and density functional theory (DFT) with the PBE exchange correlation functional[34]. The employed Gaussian atomic orbital bases include angular momenta up to $l = 2$ (d-orbitals). We augment the training data by adding rotated geometries and correspondingly rotated Hamiltonian and overlap matrices to learn the correct rotational symmetries (see Methods section). Detailed model and training settings for each data set are listed in Supplementary Table 1.

As Supplementary Table 2 shows, the total energies could be predicted up to a mean absolute error below 2 meV for the molecules. The predictions show mean absolute errors below 8 meV for the Hamiltonian and below $1 \times 10^{-4}$ for the overlap matrices. We examine how these errors propagate to orbital energy and coefficients. Figure 2e shows mean absolute errors for energies of the lowest 20 molecular orbitals for ethanol reference calculations using DFT as well as HF. The errors for the DFT reference data are consistently lower. Beyond that, the occupied orbitals (1–13) are predicted with higher accuracy (<20 meV) than the virtual orbitals (~100 meV). We conjecture that the larger error for virtual orbitals arises from the fact that these are not strictly defined by the underlying data from the HF and Kohn–Sham DFT

calculations. Virtual orbitals are only defined up to an arbitrary unitary transformation. Their physical interpretation is limited and, in HF and DFT theory, they do not enter in the description of ground-state properties. For the remaining data sets, the average errors of the occupied orbitals are <10 meV for water and malondialdehyde, as well as 48 meV for uracil. This is shown in detail in Supplementary Fig. 1. The orbital coefficients are predicted with cosine similarities ≥90% (see Supplementary Fig. 2). Figure 2f depicts the predicted and reference orbital energies for the frontier MOs of ethanol (solid and dotted lines, respectively), as well as the orbital shapes derived from the coefficients. Both occupied and unoccupied energy levels are reproduced with high accuracy, including the highest occupied (HOMO) and lowest unoccupied orbitals (LUMO). This trend is also reflected in the overall shape of the orbitals. Even the slightly higher deviations in the orbital energies observed for the third and fourth unoccupied orbital only result in minor deformations. The learned covariance of molecular orbitals for rotations of a water molecule is shown in Supplementary Fig. 3.

The ML model uses about 93 million parameters to predict a large Hamiltonian matrix with >100 atomic orbitals. This size is comparable to state-of-the-art neural networks for the generation of similarly sized images[35]. Supplementary Table 6 shows the computational costs of calculating the reference data, training the network and predicting Hamiltonians. While training of SchNOrb took about 80 h, performing the required DFT reference calculations remains the bottleneck for obtaining a trained network, in particular for larger molecules. Our approach to predicting Hamiltonian matrices leads to accelerations of 2–3 orders of magnitude.

As SchNOrb learns the electronic structure of molecular systems, all chemical properties that are defined as quantum mechanical operators on the wavefunctions can be computed from the ML prediction without the need to train a separate model. We investigate this feature by directly calculating electronic dipole and quadrupole moments from the orbital coefficients predicted by SchNOrb, as well as the HF total energies for the ethanol molecule. The corresponding mean absolute errors are reported in Supplementary Tables 4 and 5. The calculation of energies and forces from the coefficients requires the evaluation of the core Hamiltonian (HF) or exchange correlation terms (DFT), for which we currently resort to the ORCA code. To avoid this computational overhead and obtain highly accurate predictions for molecular dynamics simulations with mean absolute error below 1 meV, we predict energies and forces directly as a sum of atomic contributions[31]. Regarding the electrostatic moments, excellent agreement with the electronic structure reference is observed for the majority of molecules (<0.054 D for dipoles and <0.058 D Å for quadrupoles). The only deviation from this trend is observed for uracil, where a loss function minimising only the errors of Hamiltonian and overlap matrices is too limited. The dipole moment depends strongly on the molecular electron density derived from the orbital coefficients, which are never learned directly.

Beyond that, we have studied the prediction accuracy for ethanol when using the larger def2-tzvp basis set which includes f-orbitals ($l = 3$). While the predictions of the Hamiltonian and overlap matrices remain remarkably accurate with 8.3 meV and $10^{-6}$, respectively, the derived properties exhibit large errors, e.g., an MAE of 0.4775 eV for the orbital energies. For large numbers of orbitals, errors in the Hamiltonian can accumulate due to the diagonalisation. This problem could be solved by improving the neural network architecture to further reduce the prediction error or introducing a density dependent term into the loss function, which will be explored in future investigations.

In this case, a similar accuracy as the other methods could in principle be reached upon the addition of more reference data

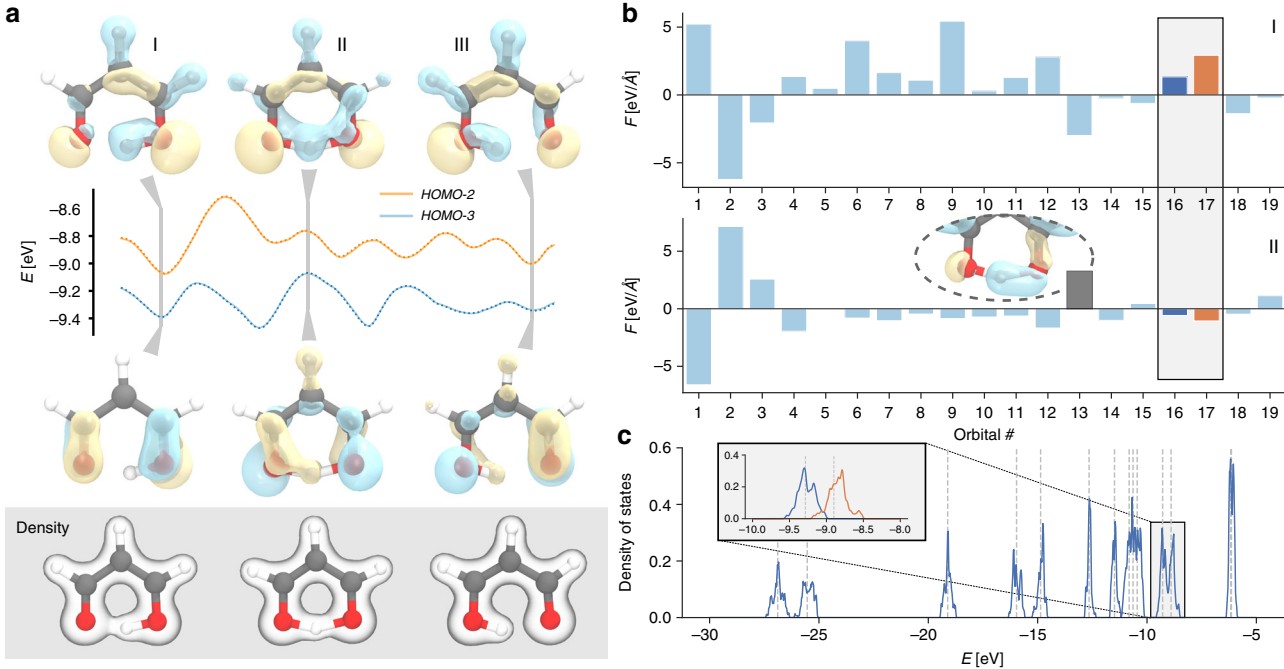

**Fig. 3** Proton transfer in malondialdehyde. **a** Excerpt of the MD trajectory showing the proton transfer, the electron density as well as the relevant MOs HOMO-2 and HOMO-3 for three configurations (I, II, III). **b** Forces exerted by the MOs on the transferred proton for configurations I and II. **c** Density of states broadened across the proton transfer trajectory. MO energies of the equilibrium structure are indicated by grey dashed lines. The inset shows a zoom of HOMO-2 and HOMO-3

points. The above results demonstrate the utility of combining a learned Hamiltonian with quantum operators. This makes it possible to access a wide range of chemical properties without the need for explicitly developing specialised neural network architectures.

**Chemical insights from electronic deep learning.** Recently, a lot of research has focused on explaining predictions of ML models[36–38] aiming both at the validation of the model[39,40] as well as the extraction of scientific insight[17,31,41]. However, these methods explain ML predictions either in terms of the input space, atom types and positions in this case, or latent features such as local chemical potentials[31,42]. In quantum chemistry however, it is more common to analyse electronic properties in terms of the MOs and properties derived from the electronic wavefunction, which are direct output quantities of the SchNOrb architecture.

Molecular orbitals encode the distribution of electrons in a molecule, thus offering direct insights into its underlying electronic structure. They form the basis for a wealth of chemical bonding analysis schemes, bridging the gap between quantum mechanics and abstract chemical concepts, such as bond orders and atomic partial charges[30]. These quantities are invaluable tools in understanding and interpreting chemical processes based on molecular reactivity and chemical bonding strength. As SchNOrb yields the MOs, we are able to apply population analysis to our ML predictions. Figure 2d shows Loewdin partial atomic charges and bond orders for the uracil molecule. Loewdin charges provide a chemically intuitive measure for the electron distribution and can e.g., aid in identifying potential nucleophilic or electrophilic reaction sites in a molecule. The negatively charged carbonyl oxygens in uracil, for example, are involved in forming RNA base pairs. The corresponding bond orders provide information on the connectivity and types of bonds between atoms. In the case of uracil, the two double bonds of the carbonyl groups are easily

recognisable (bond order 2.12 and 2.14, respectively). However, it is also possible to identify electron delocalisation effects in the pyrimidine ring, where the carbon double bond donates electron density to its neighbours. A population analysis for malondialdehyde, as well as population prediction errors for all molecules can be found in Supplementary Fig. 4 and Supplementary Table 3.

The SchNOrb architecture enables an accurate prediction of the electronic structure across molecular configuration space, which provides for rich chemical interpretation during molecular reaction dynamics. Figure 3a shows an excerpt of a molecular dynamics simulation of malondialdehyde that was driven by atomic forces predicted using SchNOrb. It depicts the proton transfer together with the relevant MOs and the electronic density. Supplementary Video 1 shows a side-by-side comparison between the predicted and reference HOMO-2 orbital during this excerpt of the trajectory. The density paints an intuitive picture of the reaction as it migrates along with the hydrogen. This exchange of electron density during proton transfer is also reflected in the orbitals. Their dynamical rearrangement indicates an alternation between single and double bonds. The latter effect is hard to recognise based on the density alone and demonstrates the wealth of information encoded in the molecular wavefunctions.

Figure 3b depicts the forces the different MOs exert onto the hydrogen atom exchanged during the proton transfer. All forces are projected onto the reaction coordinate, where positive values correspond to a force driving the proton towards the product state. In the initial configuration I, most forces lead to attraction of the hydrogen atom to the right oxygen. In the intermediate configuration II, orbital rearrangement results in a situation where the majority of orbital force contributions on the hydrogen atom become minimal, representing mostly non-bonding character between oxygens and hydrogen. One exception is MO 13, depicted in the inset of Fig. 3b. Due to a minor deviation from a symmetric O–H–O arrangement, the orbital represents a one-

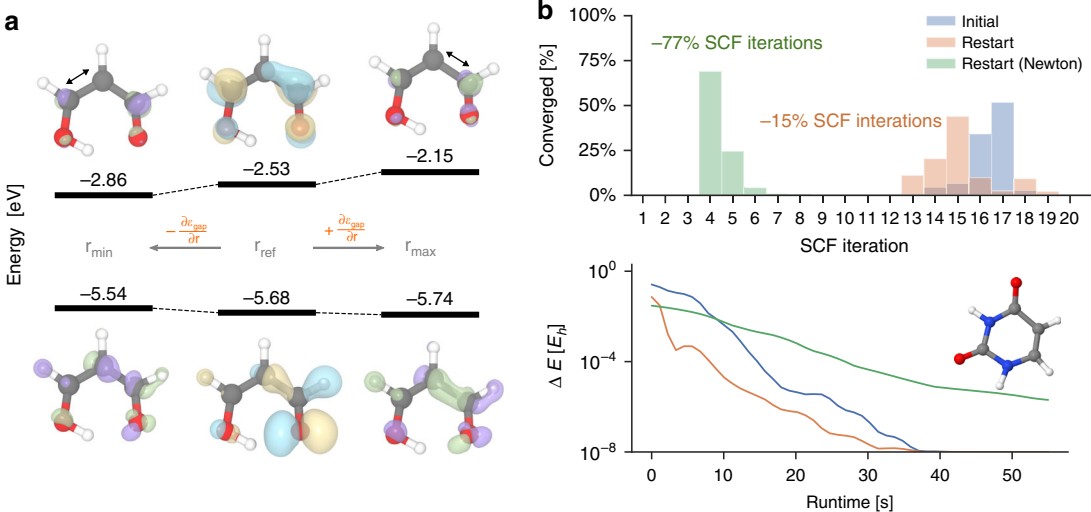

**Fig. 4** Applications of SchNOrb. **a** Optimisation of the HOMO-LUMO gap. HOMO and LUMO with energy levels are shown for a randomly drawn configuration of the malonaldehyde dataset (centre) as well as for configurations that were obtained from minimising or maximising the HOMO-LUMO gap prediction using SchNOrb (left and right, respectively). For the optimised configurations, the difference of the orbitals are shown in green (increase) and violet (decrease). The dominant geometrical change is indicated by the black arrows. **b** The predicted MO coefficients for the uracil configurations from the test set are used as a wavefunction guess to obtain accurate solutions from DFT at a reduced number of self-consistent-field (SCF) iterations. This reduces the required SCF iterations by an average of 77% using a Newton solver. In terms of runtime, it is more efficient to use SOSCF, even though this saves only 15% of iterations for uracil

sided O–H bond, exerting forces that promote the reaction. The intrinsic fluctuations during the proton transfer molecular dynamics are captured by the MOs as can be seen in Fig. 3c. This shows the distribution of orbital energies encountered during the reaction. As would be expected, both HOMO-2 and HOMO-3 (inset, orange and blue respectively), which strongly participate in the proton transfer, show significantly broadened peaks due to strong energy variations in the dynamics. This example nicely shows the chemically intuitive interpretation that can be obtained by the electronic structure prediction of SchNOrb.

**Deep learning-enhanced quantum chemistry.** An essential paradigm of chemistry is that the molecular structure defines chemical properties. Inverse chemical design turns this paradigm on its head by enabling property-driven chemical structure exploration. The SchNOrb framework constitutes a suitable tool to enable inverse chemical design due to its analytic representation of electronic structure in terms of the atomic positions. We can therefore obtain analytic derivatives with respect to the atomic positions, which provide the ability to optimise electronic properties. Figure 4a shows the minimisation and maximisation of the HOMO-LUMO gap $\epsilon_{gap}$ of malondialdehyde as an example. We perform gradient descent and ascent from a randomly selected configuration $\mathbf{r}_{ref}$ until convergence at $\mathbf{r}_{min}$ and $\mathbf{r}_{max}$, respectively. We are able to identify structures which minimise and maximise the gap from its initial 3.15 to 2.68 eV at $\mathbf{r}_{min}$ and 3.59 eV at $\mathbf{r}_{max}$. While in this proof of concept these changes were predominantly caused by local deformations in the carbon–carbon bonds indicated in Fig. 4a, they present an encouraging prospect how electronic surrogate models such as SchNOrb can contribute to computational chemical design using more sophisticated optimisation methods, such as alchemical derivatives[43] or reinforcement learning[44].

ML applications for electronic structure methods have usually been one-directional, i.e., ML models are trained to predict the outputs of calculations. On the other hand, models in the spirit of

Fig. 1b, such as SchNOrb, offer the prospect of providing a deeper integration with quantum chemistry methods by substituting parts of the electronic structure calculation. SchNOrb directly predicts wavefunctions based on quantum chemistry data, which in turn, can serve as input for further quantum chemical calculations. For example, in the context of HF or DFT calculations, the relevant equations are solved via a self-consistent field approach (SCF) that determines a set of MOs. The convergence with respect to SCF iteration steps largely determines the computational speed of an electronic structure calculation and strongly depends on the quality of the initialisation for the wavefunction. The coefficients predicted by SchNOrb can serve as such an initialisation of SCF calculations. To this end, we generated wavefunction files for the ORCA quantum chemistry package[45] from the predicted SchNOrb coefficients, which were then used to initialise SCF calculations. Figure 4b depicts the SCF convergence for three sets of computations on the uracil molecule: using the standard initialisation techniques of quantum chemistry codes, and the SchNorb coefficients with or without a second order solver. Nominally, only small improvements are observed using SchNorb coefficients in combination with a conventional SCF solver. This is due to the various strategies employed in electronic structure codes in order to provide a numerically robust SCF procedure. By performing SCF calculations with a second order solver, which would not converge using a less accurate starting point than our SchNorb MO coefficients, the efficiency of our combined ML and second order SCF approach becomes apparent. Convergence is obtained in only a fraction of the original iterations, reducing the number of cycles by ~77%. Similarly, Supplementary Fig. 5 shows the reduction of SCF iteration by ~73% for malondialdehyde. However, since second-order optimisation steps are more costly, it is more time-efficient to perform conventional SOSCF which reduces the convergence time by 13% and 16% for uracil and malondialdehyde, respectively.

It should be noted, that this combined approach does not introduce any approximations into the electronic structure method itself and yields exactly the same results as the full

computation. Another example of integration of the SchNOrb deep learning framework with quantum chemistry, as shown in Fig. 1b, is the use of predicted wavefunctions and MO energies based on Hartree–Fock as starting point for post-Hartree–Fock correlation methods such as Møller–Plesset perturbation theory (MP2). Supplementary Table 5 presents the mean absolute error of an MP2 calculation for ethanol based on wavefunctions predicted from SchNOrb. The associated prediction error for the test set is 83 meV. Compared to the overall HF and MP2 energy, the relative error of SchNOrb amounts to 0.01% and 0.06%, respectively. For the MP2 correlation energy, we observe a deviation of 17%, the reason of which is inclusion of virtual orbitals in the calculation of the MP2 integrals. However, even in this case, the total error only amounts to a deviation of 93 meV.

## Discussion

The SchNOrb framework provides an analytical expression for the electronic wavefunctions in a local atomic orbital representation as a function of molecular composition and atom positions. While previous approaches have predicted Hamiltonians of single-species bulk materials in a small basis set for limited geometric deformations[28], SchNorb has been shown to enable the accurate predictions of molecular Hamiltonians in a basis of >100 atomic orbitals up to angular momentum $l = 2$ for a much larger configuration space obtained from molecular dynamics simulations. As a consequence, the model provides access to atomic derivatives of wavefunctions, which include molecular orbital energy derivatives, Hamiltonian derivatives, which can be used to approximate nonadiabatic couplings[46], as well as higher order derivatives that describe the electron-nuclear response of the molecule. Thus, the SchNOrb framework preserves the benefits of interatomic potentials while enabling access to the electronic structure as predicted by quantum chemistry methods.

SchNOrb opens up completely new applications to ML-enhanced molecular simulation. This includes the construction of interatomic potentials with electronic properties that can facilitate efficient photochemical simulations during surface hopping trajectory dynamics or Ehrenfest-type mean-field simulations, but also enables the development of new ML-enhanced approaches to inverse molecular design via electronic property optimisation.

The SchNOrb neural network architecture demonstrates that an accurate prediction of electronic structure is feasible. However, with increasing number of atomic orbitals, the diagonalisation of the Hamiltonian leads to the accumulation of prediction errors and becomes the bottleneck of the prediction. Thus, for larger molecules and basis sets, the accuracy of SchNOrb will have to be further improved. More research on the architecture is also required in order to reduce the required amount of training data and parameters, e.g., by adding more prior knowledge to the model. An import step into this direction is to encode the full rotational symmetries of the basis into the architecture, replacing the current data augmentation scheme. Alternatively, on the basis of the SchNOrb framework, intelligent preprocessing of quantum chemistry data in the form of effective Hamiltonians or optimised minimal basis representations[47] can be developed in the future. Such preprocessing will also pave the way towards the prediction of the electronic structure based on post-HF correlated wavefunction methods and post-DFT quasiparticle methods.

This work serves as a first proof of principle that direct ML models of electronic structure based on quantum chemistry can be constructed and used to enhance further quantum chemistry calculations. We have presented an immediate consequence of this by reducing the number of DFT-SCF iterations with

wavefunctions predicted via SchNOrb. The presented model delivers derived electronic properties that can be formulated as quantum mechanical expectation values. This provides an important step towards a full integration of ML and quantum chemistry into the scientific discovery cycle.

## Methods

**Reference data.** Reference configurations were sampled at random from the MD17 dataset[7] for each molecule. The number of selected configurations per molecule is given in Supplementary Table 1. All reference calculations were carried out with the ORCA quantum chemistry code[45] using the def2-SVP basis set[48]. Integration grid levels of 4 and 5 were employed during SCF iterations and the final computation of properties, respectively. Unless stated otherwise, the default ORCA SCF procedure was used, which is based on the Pulay method[49]. For the remaining cases, the Newton–Raphson procedure implemented in ORCA was employed as a second order SCF solver. SCF convergence criteria were set to VeryTight. DFT calculations were carried out using the PBE functional[34]. For ethanol, additional HF computations were performed.

Molecular dynamics simulations for malondialdehyde were carried out with SchNetPack[50]. The equations of motions were integrated using a timestep of 0.5 fs. Simulation temperatures were kept at 300 K with a Langevin thermostat[51] employing a time constant of 100 fs. Trajectories were propagated for a total of 50 ps, of which the first 10 ps were discarded.

**Details on the neural network architecture.** In the following we describe the neural network depicted in Fig. 2 in detail. We use shifted softplus activation functions

$$ssp(x) = \ln\left(\frac{1}{2}e^x + \frac{1}{2}\right) \tag{7}$$

throughout the architecture. Linear layers are written as

$$\text{linear}(\mathbf{x}) = \mathbf{W}^{\mathsf{T}}\mathbf{x} + \mathbf{b} \tag{8}$$

with input $\mathbf{x} \in \mathbb{R}^{n_{in}}$, weights $\mathbf{W} \in \mathbb{R}^{n_{in} \times n_{out}}$ and bias $\mathbf{b} \in \mathbb{R}^{n_{out}}$. Fully-connected neural networks with one hidden layer are written as

$$\text{mlp}(\mathbf{x}) = \mathbf{W}_2^{\mathsf{T}} ssp(\mathbf{W}_1^{\mathsf{T}}\mathbf{x} + \mathbf{b}_1) + \mathbf{b}_2 \tag{9}$$

with weights $\mathbf{W}_1 \in \mathbb{R}^{n_{in} \times n_{hidden}}$ and $\mathbf{W}_2 \in \mathbb{R}^{n_{hidden} \times n_{out}}$ and biases $\mathbf{b}_1, \mathbf{b}_2$ accordingly. Model parameters are shared within layers across atoms and interactions, but never across layers. We omit layer indices for clarity.

The representations of atomic environments are constructed with the neural network structure as in SchNet. In the following, we summarise this first part of the model. For further details, please refer to Schütt et al.[32]. First, each atom is assigned an initial element-specific embedding

$$\mathbf{x}_i^0 = \mathbf{a}_{Z_i} \in \mathbb{R}^B, \tag{10}$$

where $Z_i$ is the nuclear charge and $B$ is the number of atom-wise features. In this work, we use $B = 1000$ for all models. The representations are refined using SchNet interaction layers (Fig. 2b). The main component is a continuous-filter convolutional layer (cfconv)

$$\text{cfconv}((\mathbf{x}_1, r_1), \ldots, (\mathbf{x}_n, r_n)) = \left[\sum_{j \neq i} \mathbf{x}_j \circ \mathbf{W}_{\text{filter}}(r_{ij})\right]_{i=1 \ldots n} \tag{11}$$

which takes a spatial filter $\mathbf{W}_{\text{filter}} : \mathbb{R} \to \mathbb{R}^B$

$$\mathbf{W}_{\text{filter}}(r_{ij}) = \text{mlp}(\mathbf{g}(r_{ij}))\, f_{\text{cutoff}}(r_{ij}) \tag{12}$$

with

$$\mathbf{g}(r_{ij}) = \left[\exp(-\gamma(r_{ij} - k\Delta\mu)^2)\right]_{0 \leq k \leq r_c/\Delta\mu} \tag{13}$$

$$f_{\text{cutoff}}(r_{ij}) = \begin{pmatrix} 0.5 \times \left[1 + \cos\left(\frac{\pi r}{r_c}\right)\right] & r < r_c \\ 0 & r \geq r_c \end{pmatrix}, \tag{14}$$

where $r_c$ is the cutoff radius and $\Delta\mu$ is the grid spacing of the radial basis function expansion of interatomic distance $r_{ij}$. While this adds spatial information to the environment representations for each feature separately, the crosstalk between features is performed atom-wise by fully-connected layers (linear and $\text{mlp}_{\text{atom}}$ in Fig. 2b) to obtain the refinements $\mathbf{v}_i^t$, where $t$ is the current interaction iteration. The refined atom representations are then

$$\mathbf{x}_i^t = \mathbf{x}_i^{t-1} + \mathbf{v}_i^t. \tag{15}$$

These representations of atomic environments are employed by SchNet to predict chemical properties via atom-wise contributions. However, in order to extend this scheme to the prediction of the Hamiltonian, we need to construct representations of pair-wise environments in a second interaction phase.

The Hamiltonian matrix is of the form

$$\mathbf{H} = \begin{bmatrix} \mathbf{H}_{11} & \cdots & \mathbf{H}_{1j} & \cdots & \mathbf{H}_{1n} \\ \vdots & \ddots & \vdots & & \vdots \\ \mathbf{H}_{i1} & \cdots & \mathbf{H}_{ij} & \cdots & \mathbf{H}_{in} \\ \vdots & & \vdots & \ddots & \vdots \\ \mathbf{H}_{n1} & \cdots & \mathbf{H}_{nj} & \cdots & \mathbf{H}_{nn} \end{bmatrix} \tag{16}$$

where a matrix block $\mathbf{H}_{ij} \in \mathbb{R}^{n_{ao,i} \times n_{ao,j}}$ depends on the atoms $i, j$ within their chemical environment as well as on the choice of $n_{ao,i}$ and $n_{ao,j}$ atomic orbitals, respectively. Therefore, SchNOrb builds representations of these embedded atom pairs based on the previously constructed representations of atomic environments. This is achieved through the SchNOrb interaction module (see Fig. 2c).

First, a raw representation of atom pairs is obtained using a factorised tensor layer[31,52]:

$$\mathbf{h}_{ij}^{\lambda} = \text{ftensor}(\mathbf{x}_i^{\lambda}, \mathbf{x}_j^{\lambda}, r_{ij}) \tag{17}$$

$$= \text{ssp}(\text{linear}_2[\text{linear}_1(\mathbf{x}_i^{\lambda}) \circ \text{linear}_1(\mathbf{x}_j^{\lambda}) \circ \mathbf{W}_{\text{filter}}(r_{ij})]).$$

The layers $\text{linear}_1 : \mathbb{R}^B \mapsto \mathbb{R}^B$ map the atom representations to the factors, while the filter-generating network $\mathbf{W}_{\text{filter}} : \mathbb{R} \to \mathbb{R}^B$ is defined analogously to Eq. (12) and directly maps to the factor space. In analogy to how the SchNet interactions are used to build atomic environments, SchNOrb interactions are applied several times, where each instance further refines the atomic environments of the atom pairs with additive corrections

$$\mathbf{x}_i^{T+\lambda} = \mathbf{x}_i^{T+\lambda-1} + \mathbf{v}_i^{T+\lambda} \tag{18}$$

$$\mathbf{v}_i^{T+\lambda} = \text{mlp}_{\text{atom}}\left(\sum_{j \neq i} \mathbf{h}_{ij}^l\right) \tag{19}$$

as well as constructs pair-wise features:

$$\mathbf{p}_{ij}^{\lambda} = \mathbf{p}_{ij}^{\lambda,\text{pair}} + \sum_{m \neq i} \mathbf{p}_{mj}^{\lambda,\text{env}} + \sum_{n \neq j} \mathbf{p}_{in}^{\lambda,\text{env}} \tag{20}$$

$$\mathbf{p}_{ij}^{\lambda,\text{pair}} = \text{mlp}_{\text{pair}}(\mathbf{h}_{ij}^{\lambda}) \tag{21}$$

$$\mathbf{p}_{ij}^{\lambda,\text{env}} = \text{mlp}_{\text{env}}(\mathbf{h}_{ij}^{\lambda}) \tag{22}$$

where $\text{mlp}_{\text{pair}}$ models the direct interactions of atoms $i, j$ while $\text{mlp}_{\text{env}}$ models the interactions of the pair with neighbouring atoms. As described above, the atom pair coefficients are used to form a basis set

$$\boldsymbol{\omega}_{ij}^{\lambda} = \begin{cases} \mathbf{p}_{ij}^0 \otimes \mathbb{1}_D & \text{for } \lambda = 0 \\ \left[\mathbf{p}_{ij}^{\lambda} \otimes \dfrac{\mathbf{r}_{ij}}{\|\mathbf{r}_{ij}\|}\right]\mathbf{W} & \text{for } \lambda > 0 \end{cases}$$

where $\lambda$ corresponds to the angular momentum channel and $\mathbf{W} \in \mathbb{R}3 \times D$ are learnable parameters to project along $D$ directions. For all results in this work, we used $D = 4$. For interactions between s-orbitals, we consider the special case $\lambda = 0$ where the features along all directions are equal due to rotational invariance. At this point, $\boldsymbol{\omega}_{ij}^0$ is rotationally invariant and $\boldsymbol{\omega}_{ij}^{\lambda>0}$ is covariant. On this basis, we obtain features with higher angular momenta using:

$$\boldsymbol{\Omega}_{ij}^l = \prod_{\lambda=0}^{l} \boldsymbol{\omega}_{ij}^{\lambda} \quad \text{with } 0 \leq l \leq 2L,$$

where features $\boldsymbol{\Omega}_{ij}^l$ possess angular momentum $l$. The SchNOrb representations of atom pairs embedded in their chemical environment, that were constructed from the previously constructed SchNet atom-wise features, will serve in a next step to predict the corresponding blocks of the ground-state Hamiltonian.

Finally, we assemble the Hamiltonian and overlap matrices to be predicted. Each atom pair block is predicted from the corresponding features $\boldsymbol{\Omega}_{ij}^l$:

$$\tilde{\mathbf{H}}_{ij} = \begin{cases} \mathbf{H}_{\text{off}}\left(\left[\boldsymbol{\Omega}_{ij}^l\right]_{0 \leq l \leq 2L+1}\right) & \text{for } i \neq j \\ \mathbf{H}_{\text{on}}\left(\left[\boldsymbol{\Omega}_{im}^l\right]_{\substack{m \neq i \\ 0 \leq l \leq 2L+1}}\right) & \text{for } i = j \end{cases},$$

where we restrict the network to linear layers in order to conserve the angular momenta:

$$\mathbf{H}_{\text{off}}(\cdot) = \sum_l \text{linear}_{\text{off}}^l\left(\boldsymbol{\Omega}_{ij}^l\right)[:n_{ao,i}, :n_{ao,j}]$$

$$\mathbf{H}_{\text{on}}(\cdot) = \sum_{j,l} \text{linear}_{\text{on}}^l\left(\boldsymbol{\Omega}_{ij}^l\right)[:n_{ao,i}, :n_{ao,i}]$$

with $\text{linear}_{\text{off}}, \text{linear}_{\text{on}} : \mathbb{R}^{2L+1 \times D} \to \mathbb{R}^{n_{ao,max} \times n_{ao,max}}$, i.e., mapping to the maximal number of atomic orbitals in the data. Then, a mask is applied to the matrix block to yield only $\tilde{\mathbf{H}}_{ij} \in \mathbb{R}^{n_{ao,i} \times n_{ao,j}}$. Finally, we symmetrise the predicted Hamiltonian:

$$\mathbf{H} = \frac{1}{2}(\tilde{\mathbf{H}} + \tilde{\mathbf{H}}^{\mathsf{T}}) \tag{23}$$

The overlap matrix is obtained similarly with blocks

$$\mathbf{S}_{\text{off}}(\cdot) = \text{linear}_{S,\text{on}}(\boldsymbol{\Omega}_{ij})[:n_{ao,i}, :n_{ao,j}]$$

$$\mathbf{S}_{ii} = \mathbf{S}_{Z_i}.$$

The prediction of the total energy is obtained analogously to SchNet as a sum over atom-wise energy contributions:

$$E = \sum_i \text{mlp}_E(\mathbf{x}_i).$$

**Data augmentation**. While SchNOrb constructs features $\boldsymbol{\Omega}_{ij}^l$ and $\boldsymbol{\Omega}_{ii}^l$ with angular momenta such that the Hamiltonian matrix can be represented as a linear combination of those, it does not encode the full rotational symmetry a priori. However, this can be learned by SchNOrb assuming the training data reflects enough rotations of a molecule. To save computing power, we reduce the amount of reference calculations by randomly rotating configurations before each training epoch using Wigner $\mathcal{D}$ rotation matrices[53]. Given a randomly sampled rotor $R$, the applied transformations are

$$\tilde{\mathbf{r}}_i = \mathcal{D}^{(1)}(R)\mathbf{r}_i \tag{24}$$

$$\tilde{\mathbf{F}}_i = \mathcal{D}^{(1)}(R)\mathbf{F}_i \tag{25}$$

$$\tilde{\mathbf{H}}_{\mu\nu} = \mathcal{D}^{(l_\mu)}(R)\mathbf{H}_{\mu\nu}\mathcal{D}^{(l_\nu)}(R) \tag{26}$$

$$\tilde{\mathbf{S}}_{\mu\nu} = \mathcal{D}^{(l_\mu)}(R)\mathbf{S}_{\mu\nu}\mathcal{D}^{(l_\nu)}(R) \tag{27}$$

for atom positions $\mathbf{r}_i$, atomic forces $\mathbf{F}_i$, Hamiltonian matrix $\mathbf{H}$, and overlap $\mathbf{S}$.

**Neural network training**. For the training, we used a combined loss to train on energies $E$, atomic forces $\mathbf{F}$, Hamiltonian $\mathbf{H}$ and overlap matrices $\mathbf{S}$ simultaneously:

$$\ell\left[(\tilde{\mathbf{H}}, \tilde{\mathbf{S}}, \tilde{E}), (\mathbf{H}, \mathbf{S}, E, \mathbf{F})\right] = \left\|\mathbf{H} - \tilde{\mathbf{H}}\right\|_F^2 + \left\|\mathbf{S} - \tilde{\mathbf{S}}\right\|_F^2 + \rho\left\|E - \tilde{E}\right\|^2$$

$$+ \frac{1 - \rho}{n_{\text{atoms}}} \sum_{i=0}^{n_{\text{atoms}}} \left\|\mathbf{F}_i - \left(-\frac{\partial \tilde{E}}{\partial \mathbf{r}_i}\right)\right\|^2 \tag{28}$$

where the variables marked with a tilde refer to the corresponding predictions and $\rho$ determines the trade-off between total energies and forces. The neural networks were trained with stochastic gradient descent using the ADAM optimiser[54]. We reduced the learning rate using a decay factor of 0.8 after $t_{\text{patience}}$ epochs without improvement of the validation loss. The training is stopped at $\text{lr} \leq 5 \times 10^{-6}$. The mini-batch sizes, patience and data set sizes are listed in Supplementary Table 1. Afterwards, the model with lowest validation error is selected for testing.

## Data availability
All datasets used in this work have been made available on http://www.quantum-machine.org/datasets.

## Code availability
All code developed in this work will be made available upon request.

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

## Acknowledgements

We gratefully acknowledge support by the Institute of Pure and Applied Mathematics (IPAM) at the University of California Los Angeles during a long program workshop. R.J.M. acknowledges funding through a UKRI Future Leaders Fellowship (MR/S016023/1). K.T.S. and K.R.M. acknowledge support by the Federal Ministry of Education and Research (BMBF) for the Berlin Center for Machine Learning (01IS18037A). This project has received funding from the European Unions Horizon 2020 research and innovation program under the Marie Skłodowska-Curie grant agreement No. 792572. Computing resources have been provided by the Scientific Computing Research Technology Platform of the University of Warwick, and the EPSRC-funded high end computing Materials Chemistry Consortium (EP/R029431/1). K.R.M. acknowledges partial financial support by the German Ministry for Education and Research (BMBF) under Grants 01IS14013A-E, 01GQ1115 and 01GQ0850; Deutsche Forschungsgesellschaft (DFG) under Grant Math+, EXC 2046/1, Project ID 390685689 and by the Technology Promotion (IITP) grant funded by the Korea government (Nos. 2017-0-00451, 2017-0-01779). Correspondence to R.J.M., K.R.M., and A.T.

## Author contributions

K.T.S. and R.J.M. proposed the approach of this work. K.T.S. conceived and implemented SchNOrb. M.G. and R.J.M. carried out the reference calculations. K.T.S. and M.G. performed the molecular dynamics simulations and prepared the figures. K.T.S., M.G., and R.J.M. designed the analyses and wrote the paper. K.T.S., M.G., A.T., K.R.M., and R.J.M. discussed results and commented on the manuscript.

## Competing interests

The authors declare no competing interests.

**Additional information**

