## [Peer Review File · Nature Communications]

Reviewers' comments:

Reviewer #1 (Remarks to the Author):

Maurer et al. in their manuscript titled 'Unifying machine learning and quantum chemistry - a deep neural network for molecular wavefunctions' develop their ML approach for learning QM Hamiltonian matrix elements, energies, and forces of specific small molecular systems. They also suggest to use trained ML models for accelerating SCF convergence. It is a good, solid work with some interesting, although oftentimes trivial or not very new, insights. My comments on specific topics are below:

* The idea of using ML for learning wavefunction is not that new: Sugawara suggested using NN for solving the Schrodinger equation back in 2001 in Computer Physics Communications (vol. 140, p. 366), albeit for simple model potentials. The authors also cite ref. 21, where ML was applied in a very similar way to the authors' manuscript, just with a different ML method. Tucker Carrington Jr. has used NN to solve the vibrational Schrodinger equation. This all should be mentioned clearer and in a more prominent place in Introduction, because now this information is either missing or hidden somewhere in the discussion. What seems to be new is the visualization of ML orbitals and SCF acceleration.

* The authors' claim of "full access to the electronic structure at force-field-like efficiency" is exaggerated. First of all, the authors do not really explicitly mention how they obtain orbital coefficients and eigenvalues, but I assume they need perform ML-Fock matrix diagonalization, which is rather slow. In addition, ML should be slower than molecular mechanics as it need evaluate many more terms.

* Fig. 1b is a nice concept, but is misleading in the context of the manuscript, because the authors actually never use this concept. They do train ML on QM reference energies and forces, i.e. these properties are part of the training set. In addition, they 'predict the total energy separately as a sum over atom-wise energy contributions, in analogy with the conventional SchNet treatment²⁴,' i.e. not directly from the ML wavefunction.

* All this discussion in paragraph starting with 'ML applications in chemistry have traditionally been one-directional, i.e. ML models are built on quantum chemistry data.' tries to convince the reader that the manuscript presents something very unique, not one-directional. In reality, their NN models are also built on QC data, and other works in the literature as learning-on-the-fly ML dynamics

(numerous groups) or geometry optimization accelerated by ML (works by Alexander Denzel and Johannes Kästner; Ove Christiansen) use similar concept of "deeper integration".

* The authors should provide more technical details how they integrated ML and QC for SCF: did they made modifications to the ORCA code?

* Technical information about training/testing is too sparse. They need be clearer how exactly they generated train/validation/test sets.

* In the loss function, they never explained what 'ro' in eq. 28 is.

* Their approach seem to be extremely complex, the authors should provide more details on: the total number of parameters to fit for each system, how much time it took for training and on what computer architecture.

* Code and data availability should be discussed.

Reviewer #2 (Remarks to the Author):

This is an interesting paper that is expected to play an important role in the effort aimed at exploiting machine learning for chemistry through quantum chemistry. The authors propose and demonstrate a new neural network architecture to predict the wave functions, from which observables can be computed. This makes the method more general than the previously proposed approaches. I recommend the paper for publication.

The only suggestion I would have for the authors is to include - perhaps as a separate section - an in-depth analysis of the prediction errors. Is there a limit on the accuracy that can be achieved? Can the authors present a broad statistic of errors by analyzing a large number of molecules? Can some properties be predicted more accurately than others, and if yes, again is there a fundamental accuracy limit? Can the present neural network be improved to increase the accuracy?

Some of these questions have already been addressed, some in the supplementary material, but I think the paper would benefit for a clear section addressing this important issue to the general reader, because this will clearly illustrate to the readers the value of this effort for chemical discovery. Basically, I am - and the reader will be - wondering if machine learning will/can ultimately replace quantum chemistry for applications such as in organic synthesis and drug design.

Reviewer #3 (Remarks to the Author):

Schütt et al. present an extension of their deep tensor neural network SchNet method that can predict the molecular electronic wave function and, consequently, energies, molecular orbitals, and properties. Their approach to represent the electronic Hamiltonian in a local basis representation is remarkable. The authors demonstrate the applicability of the new method (SchNOrb) in three examples: intramolecular proton transfer of malondialdehyde, inverse optimization of HOMO-LUMO gaps, and speed-up of the SCF procedure. Each of these three cases has their strengths and weaknesses, which are discussed here:

- It is very exciting to see that a ML method can predict the MOs of HF/DFT. However, all results have been obtained with a small basis set (def2-SVP). How would the method behave if a more realistic basis set (eg. def2-TZVP) is used? Does SchNOrb require more data points in order to achieve the same accuracy?
- It seems arbitrary that errors from DFT are constantly lower than HF. I would expect that SchNOrb would have comparable errors for both DFT and HF; the ML model is trained on DFT in order to predict DFT wave functions, and, similarly, trained on HF to predict HF wave functions. Or is this not the case? Is the accuracy of the ML method also functional-dependent?

- The authors claim that the method provides “chemical insights” but they show this by considering a very simple case (malondialdehyde). It is not clear from Figure 3c how such chemical insights can be extracted. Many orbitals experience a broadening in the DOS. For a textbook example such as malondialdehyde, it is clear that such information can be correlated with the proton transfer. Can we extract the same information from a more complex case that has hundreds of orbitals?

- In addition, the method can offer “chemical insights” in the same manner as DFT can, therefore there is no novelty here. It only speeds-up the DFT calculation (that’s novel). Thus, if DFT fails, then SchNOrb will fail.

- The predicted dipole moment of one of the five cases examined (uracil) is off by 1.2762D (uracil’s dipole moment is 4.428D at the B3LYP/def2-TZVPP level – from cccbdb.nist.gov). The explanation that uracil has a delocalized π -system does not seem convincing for a method that claims to predict accurately the electronic wave function.

- SchNOrb offers a better starting guess for SCF than the standard extended Hückel method which is used in conventional SCF implementations. For uracil with Newton steps, the reduction of iterations is ~77%, but what about the exact computational time? Newton-Raphson steps are ~2 times slower than DIIS. What is the time needed for training the deep network? This should also be added in the efficiency of using a SchNOrb-predicted wave function as initial guess for SCF. Do you expect similar speed-ups for a triple-zeta basis?

- How many data points (structures) are needed for training a deep network, for example uracil? This is not clear since the authors mention that SchNOrb still suffers from the lack of rotational invariance and additional structures should be included in order to reflect the rotations of the molecules. It is also not clear if individual networks were trained per molecule or just one with data from all the molecules considered in this study.

- Overall, the manuscript is well-written. The authors provide adequate information on the computational procedure that was followed, and the Supplementary material includes key results. However, they present the “unification” of ML with quantum chemistry (QC) as novel, which is not the case. Recent publications of methods that interface ML/QC are omitted in the introduction, eg.

- Smith et al., Nat. Commun. 2903,

- Welborn et al., J. Chem. Theory Comput., 14, 4772,

- Townsend et al., J. Phys. Chem. Lett. 10, 4129

Reply to Reviewer 1:

Maurer et al. in their manuscript titled 'Unifying machine learning and quantum chemistry - a deep neural network for molecular wavefunctions' develop their ML approach for learning QM Hamiltonian matrix elements, energies, and forces of specific small molecular systems. They also suggest to use trained ML models for accelerating SCF convergence. It is a good, solid work with some interesting, although oftentimes trivial or not very new, insights. My comments on specific topics are below:

**** The idea of using ML for learning wavefunction is not that new: Sugawara suggested using NN for solving the Schrodinger equation back in 2001 in Computer Physics Communications (vol. 140, p. 366), albeit for simple model potentials. The authors also cite ref. 21, where ML was applied in a very similar way to the authors' manuscript, just with a different ML method. Tucker Carrington Jr. has used NN to solve the vibrational Schrodinger equation. This all should be mentioned clearer and in a more prominent place in Introduction, because now this information is either missing or hidden somewhere in the discussion. What seems to be new is the visualization of ML orbitals and SCF acceleration.***

The referee is correct in their assessment that previous works have attempted to use ML to represent wave functions. Mentioned are works by Sugawara et al. and by Carrington et al., both of which represent attempts to use neural networks as basis set representation to span the discrete Hilbert space of wave functions and to solve the respective electronic and nuclear Schrodinger equations of simple systems. These are important works and noteworthy. We have therefore included them in the current manuscript version.

However, in this manuscript, we do not use ML to develop a basis representation to solve quantum mechanical equations, as these above works have attempted, but we use it to develop a completely general analytical parameterization of solutions of quantum mechanical methods represented within the traditional atom-centred basis functions used in quantum chemistry. In doing so, we achieve two important aspects: (A) We report a ML-based parametrization that is capable of fully capturing the rotational equivariance properties of atomic orbital basis functions up to arbitrary angular momentum, (B) by using an established basis expansion in quantum chemistry, we are able to seamlessly reintroduce ML predicted wave functions (and expectation values thereof) into established quantum chemical software and, thereby, achieve a previously underexplored synergy between the two. Both (A) and (B) have, to the best of our knowledge, not been reported before.

It is important to stress that, particularly point (A), has not been sufficiently addressed by Ref. 21 (Ref. 27 in the updated version), as this manuscript only reports a prediction of interaction integrals of up to four rotationally invariant s-orbitals and p-orbitals in single-species bulk. Rotational equivariance of the predicted wave functions is not enforced or learned in Ref 21. In contrast, our approach is shown to predict Hamiltonians of molecules with up to four atom

species and more than 100 basis functions up to d-orbitals (results with f-orbitals have been added in the updated ms). Since the molecular geometries may rotate during the MD simulation, rotational equivariance of the model is a crucial property of our model.

We understand that the previous version of the manuscript has not sufficiently stressed novelty with respect to previous works and we have amended the results and discussion sections accordingly.

On page 1, we have included additional notes and references to highlight previous works in this field. On pages 1 and 7, we have included comments to emphasize further the generality of our model compared to Ref. 21 (now Ref 27).

**** The authors' claim of "full access to the electronic structure at force-field-like efficiency" is exaggerated. First of all, the authors do not really explicitly mention how they obtain orbital coefficients and eigenvalues, but I assume they need perform ML-Fock matrix diagonalization, which is rather slow. In addition, ML should be slower than molecular mechanics as it need evaluate many more terms.***

Despite the fact that our method achieves several orders of magnitude computational speed-up and improved scaling properties when evaluating the wave function compared to quantum chemical methods, e.g. an SCF-HF calculation of the Fock matrix of the Uracil molecule on a single CPU takes 88 s, with SchNOrb it takes 2 s on a CPU and 50 ms on a GPU. These runtimes could be further reduced using model compression techniques. We agree that the real-time performance of our deep learning model lags behind force fields based on simple few-body interaction potentials. We have adjusted this claim in the present version of the manuscript.

We have also included a more detailed description of how wave function coefficients are determined, which indeed at the moment is handled via matrix diagonalisation of the highly compact Hamiltonian and overlap matrices, which represent the main output of our model. The Fock matrix diagonalisation is currently not a bottleneck of the evaluation (< 1 ms), but could become one with larger molecules and basis sets. This however, represents a shift away from the much more severe computational bottleneck of conventional quantum chemical calculations, namely the repeated Fock matrix construction and diagonalization in the SCF procedure, which is handled by our deep learning model.

Adjustments to paragraph 1 on page 2 regarding the discussion of computational speed-up have been made. On page 4, we added above discussion of the computational cost of diagonalization to the ms on page and explain how orbitals and eigenvalues are evaluated.

**** Fig. 1b is a nice concept, but is misleading in the context of the manuscript, because the authors actually never use this concept. They do train ML on QM reference energies and forces, i.e. these properties are part of the training set. In addition, they 'predict the total energy separately as a sum over atom-wise energy contributions, in analogy with the conventional SchNet treatment²⁴,' i.e. not directly from the ML wavefunction.***

We disagree with this statement. Figure 1b reflects the idea of using wave functions predicted from ML as input to evaluate QM properties, such as energies and quadrupole moments. We do indeed use this concept and show the corresponding results already in the original manuscript version. We, however, acknowledge that this was not prominently displayed in the previous version of the manuscript. Tables S4 and S5 in the original manuscript report Hartree-Fock total energies, quadrupole moments, and MP2 energies which are directly calculated solely from the SchNorb wave functions by evaluation of the respective integrals. This clearly uses the concept reflected in Figure 1b. Yet, for the prediction of total energies, we have, in the end, chosen to separately predict the energy as a sum over atom-wise energies in the usual SchNet way, as this provides a higher level of error control and enables to seamlessly predict energies from wavefunction methods and Density Functional Theory (rather than only from wavefunction methods).

On page 4 and 7, we have clarified the description of the results to better reflect our use of the ML/QM unifying capabilities of the SchNorb model and our rationale for predicting total energies independently.

**** All this discussion in paragraph starting with 'ML applications in chemistry have traditionally been one-directional, i.e. ML models are built on quantum chemistry data.' tries to convince the reader that the manuscript presents something very unique, not one-directional. In reality, their NN models are also built on QC data, and other works in the literature as learning-on-the-fly ML dynamics (numerous groups) or geometry optimization accelerated by ML (works by Alexander Denzel and Johannes Kästner; Ove Christiansen) use similar concept of "deeper integration".***

We agree with the referee that there have been many recent excellent efforts to integrate ML concepts into structure prediction and dynamics methods, however, these do not directly relate to solving the electronic structure problem.

In this manuscript, we present a deep learning model that predicts wave functions built from quantum chemical methods to feed into other quantum chemical electronic structure methods. We did not state that this is a unique effort, but we believe that it adds a new dimension to electronic structure theory with significant implications for future method development work.

The new manuscript version has been modified to make this distinction clearer and to better reflect recent related efforts in the field.

We have reformulated the mentioned paragraph on page 6 and added discussion of related work in the introduction on page 1 to better reflect the implications of our work in the context of other recent efforts to integrate concepts of ML in quantum chemistry.

**** The authors should provide more technical details how they integrated ML and QC for SCF: did they made modifications to the ORCA code?***

We use an unmodified version of ORCA. The integration of ORCA and SchNOorb was done by generating ORCA wavefunction restart files (*.gbw) from the predicted SchNOorb coefficients. These wave functions were then used to calculate observables with ORCA. It is important to stress that we have used ORCA only as quantum chemical integral engine to directly calculate all reported derived observables without solving SCF equations, with the exception of Figure 4b where we demonstrate the utility of ML predicted wave functions as initial guesses for SCF calculations.

We added a corresponding statement to Section II E of the ms.

**** Technical information about training/testing is too sparse. They need be clearer how exactly they generated train/validation/test sets.***

All reference datasets consist of molecular configurations selected at random from the MD17 database, which contains molecular dynamics trajectories of different organic molecules. For each of these structures, the desired properties were then recomputed using the target electronic structure reference method. We updated the method section of the ms to clarify this.

We added information on the sampling of the reference data to the Methods section and referred to Table S1.

**** In the loss function, they never explained what 'ro' in eq. 28 is.***

Rho determines the trade-off between total energy and force error in the loss. We added this information to the ms.

**** Their approach seem to be extremely complex, the authors should provide more details on: the total number of parameters to fit for each system, how much time it took for training and on what computer architecture.***

We appreciate the referees suggestions and have added information about the size of the model as well as computational cost on page 4 as well as Supplemental Table S6.

*** Code and data availability should be discussed.**

We will make the code available on Github after publication as an extension to our atomistic neural network package SchNetPack. The data sets will be provided on quantum-machine.org.

Reply to Reviewer 2

This is an interesting paper that is expected to play an important role in the effort aimed at exploiting machine learning for chemistry through quantum chemistry. The authors propose and demonstrate a new neural network architecture to predict the wave functions, from which observables can be computed. This makes the method more general than the previously proposed approaches. I recommend the paper for publication.

**** The only suggestion I would have for the authors is to include - perhaps as a separate section - an in-depth analysis of the prediction errors. Is there a limit on the accuracy that can be achieved? Can the authors present a broad statistic of errors by analyzing a large number of molecules? Can some properties be predicted more accurately than others, and if yes, again is there a fundamental accuracy limit? Can the present neural network be improved to increase the accuracy? Some of these questions have already been addressed, some in the supplementary material, but I think the paper would benefit for a clear section addressing this important issue to the general reader, because this will clearly illustrate to the readers the value of this effort for chemical discovery.***

With the current data, we do not believe that there is a fundamental limitation to accuracy if a sufficiently diverse training data set is supplied to a sufficiently large model. However, we find that the model error for derived properties, such as total energies and quadrupole moments when calculated from the predicted wave functions is higher than it would be when training individual models with the same data. This reveals a general tendency for errors to compound in derived properties, which future model optimisation will need to address.

At the moment, the model is trained to correctly capture the covariant phase and amplitude changes of wave functions due to rotations by data augmentation. We are already working on an improvement to this by explicitly encoding SE(3) symmetry as, for example suggested in Ref. 26 of the current manuscript. In response to the reviewers comments, we have added a discussion of the prediction accuracy and on future improvements of the architecture on pages 7-8.

**** Basically, I am - and the reader will be - wondering if machine learning will/can ultimately replace quantum chemistry for applications such as in organic synthesis and drug design.***

We share the reviewers curiosity on this matter. It is hard to see how ML models, in their current form, can completely replace quantum chemical calculations. However, we believe that the

results presented in this manuscript strongly suggest that future quantum chemical methods will be significantly augmented by and intertwined with data-driven ML approaches to achieve predictions and simulations that are currently out of reach.

Reply to Reviewer 3

Schütt et al. present an extension of their deep tensor neural network SchNet method that can predict the molecular electronic wave function and, consequently, energies, molecular orbitals, and properties. Their approach to represent the electronic Hamiltonian in a local basis representation is remarkable. The authors demonstrate the applicability of the new method (SchNOrb) in three examples: intramolecular proton transfer of malondialdehyde, inverse optimization of HOMO-LUMO gaps, and speed-up of the SCF procedure. Each of these three cases has their strengths and weaknesses, which are discussed here:

- It is very exciting to see that a ML method can predict the MOs of HF/DFT. However, all results have been obtained with a small basis set (def2-SVP). How would the method behave if a more realistic basis set (eg. def2-TZVP) is used? Does SchNOrb require more data points in order to achieve the same accuracy?

We are happy to see that the reviewer shares our enthusiasm for these results. Their point regarding generality with respect to basis set is a fair one and we have addressed this in the revised manuscript. In the supplementary information, we now have added results for a new model trained on 25,000 data points for ethanol generated with a def2-TZVP basis (see page 5 and Table S2). With the same number of training examples, the prediction errors of the Hamiltonian matrices are only slightly higher with a mean absolute error of 8.3meV, however the accuracy of the derived properties suffers due to error accumulation in the diagonalisation of the larger matrix. We have added discussion how to increase the accuracy by improving the neural network architecture and introducing a density dependent term into the loss function to control the error distribution in future work.

- It seems arbitrary that errors from DFT are constantly lower than HF. I would expect that SchNOrb would have comparable errors for both DFT and HF; the ML model is trained on DFT in order to predict DFT wave functions, and, similarly, trained on HF to predict HF wave functions. Or is this not the case? Is the accuracy of the ML method also functional-dependent?

We train on DFT data to reproduce DFT wave functions and likewise for HF. We only have trained one model, namely Ethanol with HF data, so we do not have sufficient evidence to conclude that there is a general trend or a strong method dependence of model performance. Overall, the errors are very close in range when compared between HF and DFT. For ethanol, whereas the eigenvalues are predicted more accurately with DFT, the dipole moment is predicted more accurately with HF (see Table S4). A more detailed analysis of a potential method dependence will be the focus of future work.

- The authors claim that the method provides “chemical insights” but they show this by considering a very simple case (malondialdehyde). It is not clear from Figure 3c how such chemical insights can be extracted. Many orbitals experience a broadening in the DOS. For a textbook example such as malondialdehyde, it is clear that such information can be correlated with the proton transfer. Can we extract the same information from a more complex case that has hundreds of orbitals? In addition, the method can offer “chemical insights” in the same manner as DFT can, therefore there is no novelty here. It only speeds-up the DFT calculation (that’s novel). Thus, if DFT fails, then SchNOrb will fail.

The current layout of the SchNOrb deep learning model does not limit its application to simple molecules and the systems studied in this manuscript feature up to 132 orbitals, however, more model optimisation and factorization is planned in the future to tackle substantially larger systems. The main contribution here is indeed that the SchNOrb model establishes a parametrization of the wave functions as a function of atomic positions which allows fast prediction of electronic structure. The resulting speed-up allows to obtain large amounts of electronic structure data which can be analysed afterwards, e.g. by other machine learning and big data approaches, to extract chemical insights. Figures 3a-c show that electronic structure details can directly be extracted from the model and that the analytical parametrization of the wave function enables to disentangle which orbital changes are most strongly coupled with nuclear motion of the proton - a property which defines the orbital participation during the dynamics. A second novelty is that the parameterization allows for fast analytical derivatives of electronic structure. This opens up the possibility to explore the structure-electronic property relationship for molecules, e.g. for the proof-of-principle optimization of the HOMO-LUMO gap which could be extended to alchemical derivatives to perform inverse design in future work.

- The predicted dipole moment of one of the five cases examined (uracil) is off by 1.2762D (uracil’s dipole moment is 4.428D at the B3LYP/def2-TZVPP level – from cccbdb.nist.gov). The explanation that uracil has a delocalized π -system does not seem convincing for a method that claims to predict accurately the electronic wave function.

We have improved the analysis of the prediction error for uracil in the revised manuscript. We find that the deviation of the dipole moment for uracil, and in fact, the overall larger errors for derived electronic properties, are mainly due to the fact that the loss function has not been optimised for this task. Currently, our model focuses on the accurate prediction of the Hamiltonian and overlap matrix elements and the loss function only minimises errors directly related to that. It is important to stress that most of our predictions for derived properties are excellent, despite the fact that the learning process does not explicitly target minimization of these errors. Dipole moments, in particular, are highly dependent on the molecular density derived from the orbital coefficients, which are never learned directly. While this approach poses no problem for the smaller molecules, it reaches its limitations for uracil. In this case, the Hamiltonian errors can accumulate during diagonalisation, leading to increased errors in wave function coefficients. Nevertheless, the high accuracy achieved for water, ethanol and

malondialdehyde can be seen as a success for this proof of principle study. Moreover, it is expected that the inclusion of density based terms (e.g. density matrix) directly into the loss function will improve model performance for tasks of this nature, which will be the focus of future studies. We added a corresponding statement to section C of the ms.

- SchNOrb offers a better starting guess for SCF than the standard extended Hückel method which is used in conventional SCF implementations. For uracil with Newton steps, the reduction of iterations is ~77%, but what about the exact computational time? Newton-Raphson steps are ~2 times slower than DIIS. What is the time needed for training the deep network? This should also be added in the efficiency of using a SchNOrb-predicted wave function as initial guess for SCF. Do you expect similar speed-ups for a triple-zeta basis?

We agree that the Newton Raphson procedure is costly compared to standard SCF procedures. The speedup of Newton Raphson based restarts compared to the full quantum chemical computations is minimal (8% for malondialdehyde). However, the combination of ML restarts and NR is expected to be advantageous in cases with pathological convergence. We have also conducted new experiments with the standard ORCA SOSCF algorithm, where we disabled several of the heuristics used to improve convergence for suboptimal wavefunction guesses which impeded convergence in case of the SchNOrb guesses. With this setup, SchNOrb leads to an overall speedup of 16% for malondialdehyde and 13% for uracil. We expect that this gain in computation time can be increased further with future improvements of the SchNOrb model (see reply to 3.4) We have updated Figures 4b and S5, as well as their captions, to include timings as well as the new experiments. Pertaining to the time required to train the deep networks, we refer to reply 1.6 . Since standard guess methods like extended Hückel theory also needed initial parametrization and due to the fact that SchNOrb can be used for arbitrary configurations of a molecule once trained, it was not clear how the training time should enter the analysis. Hence, we resorted to reporting only the relative difference between the individual computations.

- How many data points (structures) are needed for training a deep network, for example uracil? This is not clear since the authors mention that SchNOrb still suffers from the lack of rotational invariance and additional structures should be included in order to reflect the rotations of the molecules. It is also not clear if individual networks were trained per molecule or just one with data from all the molecules considered in this study.

Even though SchNorb has the ability to capture the rotational equivariance, we perform data augmentation to include additional rotations of molecules in the training. In every epoch, the molecules and corresponding Hamiltonian matrices are randomly rotated using Wigner D-matrices (see Methods section). Therefore, no additional reference calculations are required to learn the rotations. On page 4 of the ms, we explain the data augmentation procedure and clarify that we train separate models for each MD trajectory. Training a common model for configurational and compositional degrees of freedom is subject to future work.

- Overall, the manuscript is well-written. The authors provide adequate information on the computational procedure that was followed, and the Supplementary material includes key results. However, they present the “unification” of ML with quantum chemistry (QC) as novel, which is not the case. Recent publications of methods that interface ML/QC are omitted in the introduction, eg.
- Smith *et al.*, *Nat. Commun.* 2903,
- Welborn *et al.*, *J. Chem. Theory Comput.*, 14, 4772,
- Townsend *et al.*, *J. Phys. Chem. Lett.* 10, 4129

We agree with the referee that there is an exciting shift towards the deeper integration of ML and electronic structure theory. The works of Welborn *et al.* and Townsend *et al.* are both closely related and predict energies/cluster amplitudes based on features derived from Hartree Fock and MP2 respectively. The latter application in particular is promising and has the potential to accelerate coupled cluster computations greatly. However, our approach aims for an integration at a more fundamental level, where we do not need to rely on electronic structure coefficients and instead directly model a central quantity in form of the Hamiltonian, which can in turn be interfaced with all the formalism available to QC. This also opens alternative avenues for future research compared to the above applications, e.g. towards effective Hamiltonians. We added the references Welborn *et al.* and Townsend *et al.* and corresponding comments to the introduction, page 1.